# Preliminary Report on Golden Langur (*Trachypithecus geei*) Winter Sleep Sites

**Kuenzang Dorji** [1,2,*] **, Lori K. Sheeran** [3,*] **, Ratan Giri** [4] **, Kathleen Barlow** [3] **, Namgay Pem Dorji** [5] **and Timothy Englund** [2]

1. Nature Study Center, Ugyen Wangchuck Institute for Conservation and Environmental Research, Wangdue 16004, Bhutan
2. College of the Sciences, Central Washington University, Ellensburg, WA 98926, USA; Timothy.Englund@cwu.edu
3. Department of Anthropology & Museum Studies, Central Washington University, Ellensburg, WA 98926, USA; BarlowK@cwu.edu
4. Langthel Park Range Office, Jigme Singye Wangchuk National Park, Trongsa 33001, Bhutan; giriratan07@gmail.com
5. Department of Environmental and Life Sciences, Sherubtse College, Royal University of Bhutan, Kanglung 13001, Bhutan; namjapemdorji635@gmail.com
* Correspondence: kdorji@uwice.gov.bt (K.D.); SheeranL@cwu.edu (L.K.S.); Tel.: +975-17642189 (K.D.)

**Abstract:** Golden langurs (*Trachypithecus geei*) in Bhutan have received little research attention in the anthropic environments where most of the population lives. We recorded group sizes and compositions and documented sleep sites for 24 golden langur groups living in a biological corridor (N = 9) and near a human settlement (N = 15) in central Bhutan. We used scan sampling to document behaviors and direct observation and camera traps to record potential predators, and we recorded occurrences of mortality, including two cases of electrocution, one case of roadkill, and one langur skull recovered from a possible leopard prey cache. Golden langur groups were on average significantly larger near human settlements (13.73 individuals) than in the biological corridor (9.55 individuals), and the adult sex ratio was greater near human settlements. The golden langurs usually slept in more than one tree, and our preliminary results indicated rare re-use of the same sleep site. Golden langurs in our study area most often slept in *Sapium insigne* trees. Sleep trees' mean DBH was 51.58 cm and the mean height was 19.37 m. We intend for our preliminary data to establish the foundation for future research on the behavior and ecology of golden langurs in Bhutan.

**Keywords:** colobine; leaf monkey; langur; sleep site

## 1. Introduction

Bhutan is a conservation stronghold for seven nonhuman primates, with perhaps the most notable taxon being the golden langur (*Trachypithecus geei*). This species is endangered [1] and for the past two review cycles has appeared on the International Union for the Conservation of Nature's top 25 most endangered primates [2]. Golden langurs are endemic to western Assam in northeast India [3] and to six districts in Bhutan at elevations ranging from 199–2600 m asl [4,5]. In India, the species can be found in degraded forest fragments with secondary forest growth [6,7]. The species' total number is estimated at ~5141 individuals in India [2]. In Bhutan, national parks provide intact, strictly protected habitats, and parks are connected by forested corridors (biological corridors (BCs)). Golden langurs also range in anthropogenically altered and impacted environments, where they can be found along roads, near farms, and near construction projects such as hydroelectric dams and quarries (human settlements (HSs)) [8]. BCs are designed to conserve meta-populations of wide-ranging species and to promote the gene flow for all taxa [9]. Sustainable development and use of natural resources, including dams constructed for hydropower, are permitted in HSs [10]. In Bhutan, it is important to study golden langurs

living in HSs because this is where the majority of the population is found [5]. Golden langurs in Bhutan total ~2439 individuals [5], a number that is considerably lower than previous estimates [11], and only 33% live in national parks [5].

Habitats and associated dispersal options influence golden langurs' group sizes, densities, and compositions [12]. Golden langurs have been observed living in uni-male/multi-female groups of 3–9 individuals, bi-male/multi-female groups of 8–15 individuals, and multi-male/multi-female groups of varying sizes [2] (Figure 1). All-male bands of two to five individuals and solo males have also been reported [2]. The uni-male/multi-female social structure is considered to be the most common and stable group formation [12,13]. In a study based at Royal Manas National Park (Zhemgang, Bhutan), researchers documented an overall average group size of 7.14 individuals, and 78% of groups observed were uni-male/multi-female [4]. Shil and colleagues [12] found significant differences in golden langur average group sizes at forest-core, forest-edge, and plantation sites in Assam, with the largest groups (average of 13.9 individuals) in plantations. In human-altered habitats, langurs live in larger groups with higher densities [12,14]. For example, we observed a group of 25 golden langurs ranging near a hydroelectric dam construction site in Trongsa district, Bhutan. Anthropic environments can alter the environment in ways that influence dispersal from birth groups and that increase mortality risks from injuries due to electrocution, car collisions, and domestic dogs [6,12].

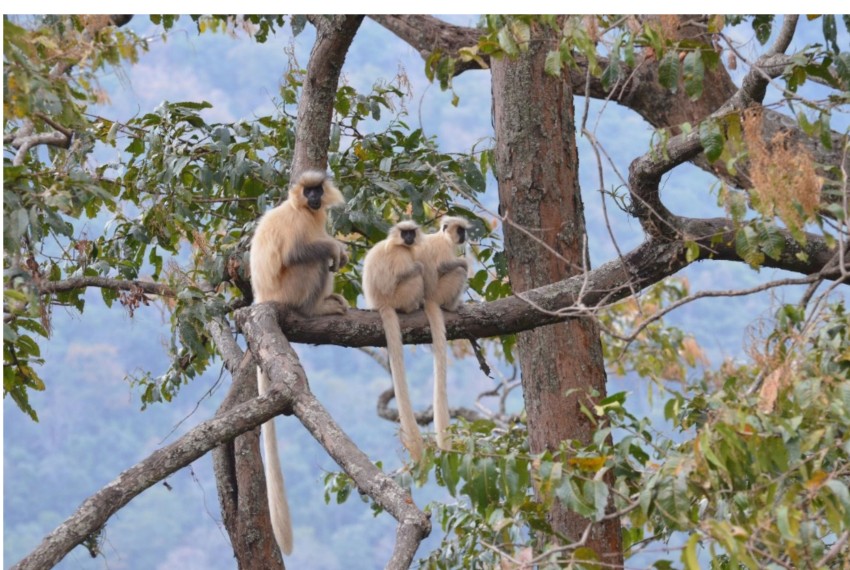

**Figure 1.** Golden langur group members basking in the sun.

Sleep sites are an important aspect of primates' habitats to study, record, and protect because nonhuman primates spend nearly half of their lives there [15–17]. Knowledge of the tree species that golden langurs prefer for sleeping will aid in the development of conservation policies that protect preferred trees across golden langurs' distribution. Qualities of sleep sites have been explained in terms of predator avoidance, food access, parasite avoidance, comfort/thermoregulation, and range/resource defense [18–21]. At sleep sites, primates may have a reduced ability to detect predators, so they may choose sleep sites with characteristics that reduce the likelihood of predation (e.g., chimpanzees (*Pan troglodytes*)) [22]. Studying golden langur–predator interactions adds to our understanding of their vulnerability to and avoidance of predators and, if langurs avoid sites with high predator density, they may seek refuge near settlements, thereby accelerating human–wildlife conflict. Golden langurs reportedly prefer to sleep in tall trees to avoid natural predators [2]. Golden langurs are described as being strictly arboreal during 99% of their active time [2], so one might expect golden langurs to sleep in the forest canopy. Aspects of sleep sites (e.g., connectivity to neighboring trees, location in the canopy, and

branching pattern) influence predators' entry to sites and primates' escape routes from them. Leopards, pythons, and raptors are among Asian primates' natural predators [23]. Chetry and colleagues [2] report leopards, wild dogs (dhole), and pythons as golden langurs' main predators. For golden langurs living near people, domestic dogs are also predators, and additional causes of mortality include electrocution on power lines, roadkill, and retaliatory killing by farmers whose crops have been damaged or destroyed by langurs [24–26].

Information on the distribution and numbers of Bhutan's golden langurs is now available, but few studies have focused on golden langur ecology, particularly the characteristics of sleep sites. We studied golden langurs ranging in Langthel subdistrict, Trongsa district, central Bhutan. Our data collection occurred between late fall and early spring, prior to the COVID-19 pandemic travel shutdown. For the first time, we provide preliminary, descriptive data from Bhutan on golden langur sleep sites and context on predation and other environmental factors that may influence sleep site selection. Based on prior observations [4,12], we hypothesized that group sizes and compositions would differ for golden langurs living in a BC and those living near HSs and predicted that group sizes would be larger and would more often include more than one male for groups near the HSs.

## 2. Materials and Methods

### 2.1. Study Site and Subjects

We collected data in Langthel sub district, central Bhutan, from 10 November 2019–30 April 2020 on golden langurs living inside a BC and near a HS (Figure 2). The BC study area totaled 154.09 km$^2$ and is administered by the Nature Conservation Division, Department of Forests and Park Services. This BC connects the Phrumshingla, Jigme Singye Wangchuk, and Royal Manas National Parks. The HS study area totaled 184.79 km$^2$ and is administered by Bumthang Forest Division. The forest type in both study areas is warm, evergreen, broadleaf forest. The subdistrict altitude ranges from 1000–4200 m asl.

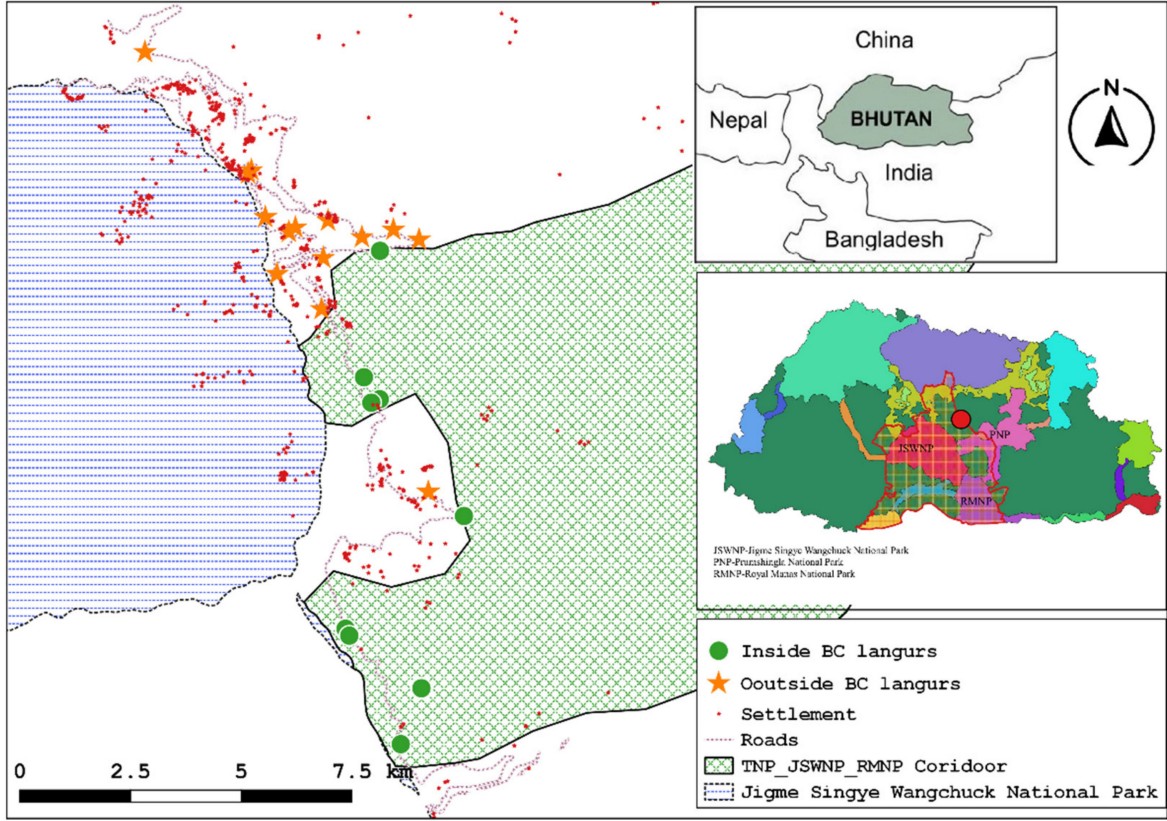

**Figure 2.** Locations of golden langur study groups in Langthel subdistrict, Trongsa district, central Bhutan.

One of us (RG) is a forester who has been based in Langthel subdistrict for 10 years and is familiar with the distribution of golden langurs in the subdistrict. Prior to in-depth observation, three authors (KD, LS, and RG) explored the study area and located 24 golden langur groups (9 in BC and 15 near HS). We assigned a unique identifier to each langur group that indicated whether it was inside a BC or near a HS. The terrain in Bhutan is rugged and steep, especially in forested regions (Figure 3). We chose these groups to observe because RG already knew their approximate ranges and locations, and we could see them along roads or by viewing them through a spotting scope positioned on the opposite mountain.

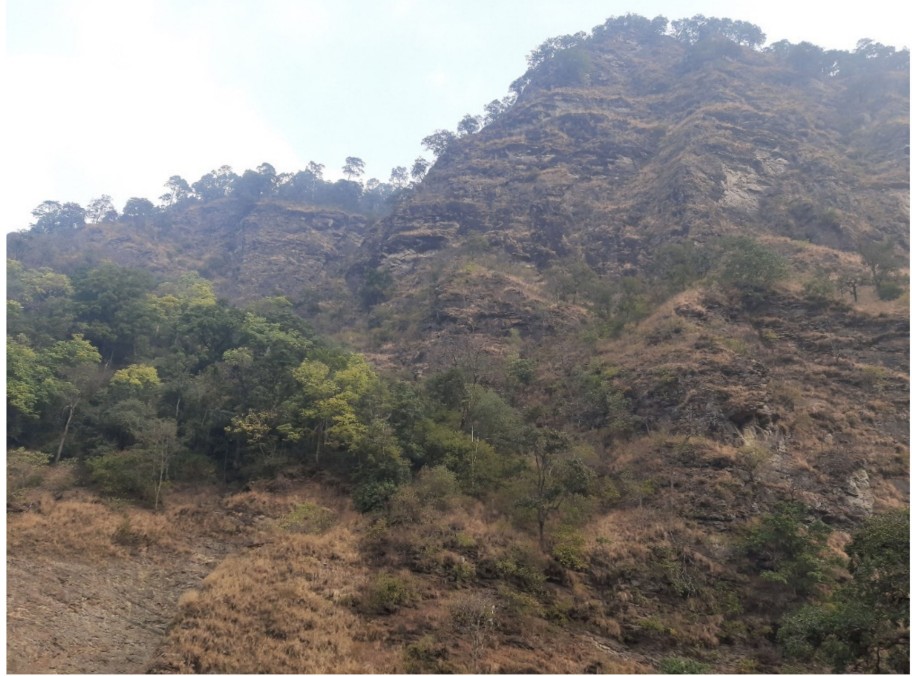

**Figure 3.** Photo showing sleep trees used by golden langurs and the steep nature of the terrain in Bhutan.

One of us (KD) instructed UWICER foresters on field techniques and equipment and, before data collection commenced, we reviewed the use of GPS and spotting scopes, identification of langur sex and age classes, GPS marking of sleep sites, collection of sleep site attributes (e.g., tree height), and use of camera traps. The first week in the field, we followed langur groups together while counting group sizes and compositions and observing behavior to ensure data were collected in the same way. Equipment used included a Vortex Viper HD 20–60 × 85 spotting scope, Celestron 8 × 42 Nature DX binoculars, a smartphone GPS and SW Maps app, Reconyx camera traps, Samsung tablets, a Sunto compass and clinometer, a Nikon Coolpix p1000, and tree diameter tape.

To document group sizes and compositions and discern groups, we followed each of the 24 golden langur groups at least twice a week from 0800 to 1100 h, which spans the time when golden langurs are usually traveling and foraging and therefore are easier to detect [27]. When we encountered a golden langur group, we counted the total number of individuals first and then attempted to classify each individual as adult male, adult female, juvenile, or infant following the age/sex classes described in [6]. We could not reliably distinguish infants and juveniles, so we combined those age classes into one category. When possible, we closely observed and recorded unique individual features such as scars and injured body parts to characterize individuals and their groups [4,5,12].

Later in the day, from 14:30 to 17:00 h, we followed a golden langur group and used scan sampling with five-minute scan intervals to collect behavioral data from the group's

members. We used this period between scans because of the large number of individuals in the group, the poor visibility in some cases, and the need to move with the langurs if they were traveling or to regain visual contact with langurs when we used a spotting scope to observe them. We noted the behavior of each group member at the start of each scan. Behaviors recorded included feeding, travel, and locations of sleep sites (these behaviors were also documented in the morning after groups were identified). Each group was also followed in the afternoon for a minimum of 2 days and a maximum of 16 days.

Using GPS, we recorded the latitude and longitude of sleep sites and feeding trees. For each sleep tree, we recorded species, girth (DBH), height, crown cover, and connectivity (Figure 4; Table 1). We classified the shape of each sleep tree into one of seven shape categories and classified the slope gradient of the ground surrounding each sleep site into one of four categories. We measured each sleep site's aspect counterclockwise from 0 to 360 degrees.

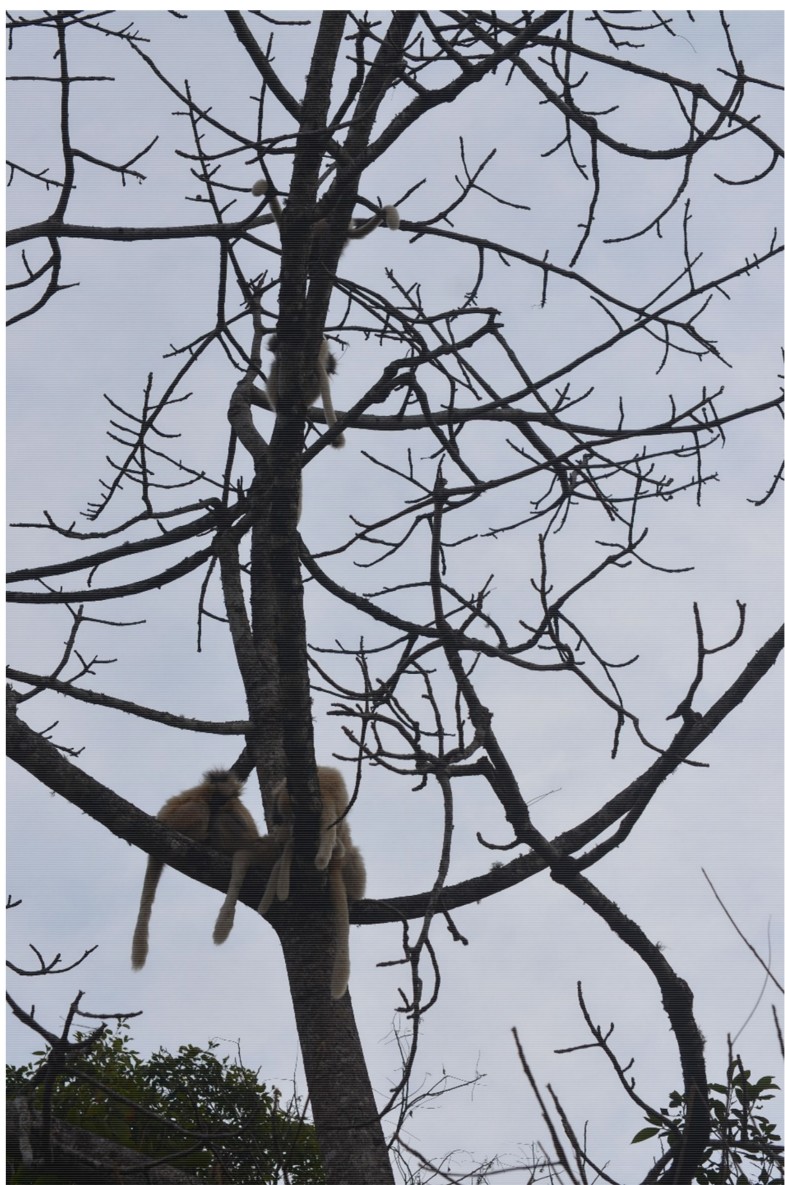

**Figure 4.** Photo showing golden langur group members in a sleep tree.

**Table 1.** Golden langur sleep tree measurements.

| Variables | Variable Description |
|---|---|
| Species | Scientific name |
| Tree type | Evergreen, deciduous, or semi-deciduous |
| DBH | Diameter of the tree at breast height, measured with tape |
| Crown cover | Sky view from tree's base, classified as low (1–49%), medium (50–84%), or dense (85–100%) |
| Height | Measured with clinometer from tree base to first branch and from first branch to tree's top |
| Tree strata | Classified as emergent, canopy, understory, or undergrowth |
| Canopy layer | Classified as understory (0–5 m), midstory (6–15 m), or overstory (>15 m) |
| Tree shape | Shape classified as columnar, pyramidal/conical, base-shaped, round/oval, spreading/oval, umbrella, or weeping |
| Tree trunk | Classified as clean/straight or branchy/crooked |
| Crown cover | Sky view from tree's base, classified as low (1–49%), medium (50–84%), or dense (85–100%) |
| Connectivity | Distance from the branch of the sleep tree to the branches of neighboring trees, classified as highly connected if within 0–10 m and lowly connected if above 10 m |
| Slope gradient | Classified as gentle (0–20%), moderate (21–30%), strong (31–50%), or steep (>50%) |

## 2.2. Survey for Predator Presence

We set camera traps to validate the predator species present. We positioned cameras near golden langurs' sleep sites and feeding ranges to maximize the probability of recording target predators. We deployed a total of 20 camera traps placed ~1 m above the ground, facing away from large objects or dense vegetation to minimize camera obstruction and false-trigger events, following previous work [28]. We walked trails that traversed golden langur feeding ranges and sleep sites from morning to dusk. We photographed, georeferenced, and recorded signs (scat, scrapes) of leopards and their ungulate prey species and signs of livestock presence, following previous work [29,30]. We used a point-count method with a fixed radius of 30 m and 15-min stops to document the presence of raptors in golden langurs' feeding ranges, following previous work [31,32]. Large-bodied, highly vocal bird species are rarely or never detected within 30 m of the observer but are more commonly detected at greater distances. We (1) recorded ad libitum evidence of raptor presence within a 30-m radius of a sleeping site to assess the intensity of natural risk of predation from raptors at this sleep location—this assessment occurred as we georeferenced the sleep site and/or encountered langurs in a sleep site; and (2) we visually and ad libitum noted raptors' presence in whole landscapes at distances >30 m away from sleep sites (for example, sightings of raptors flying over forest canopy). We also documented and georeferenced direct prey sightings and locations of golden langur carcasses.

## 2.3. Informal Interviews with Local People

RG, KD, and, on one occasion, LS conducted informal interviews with farmers. Interviews were performed as part of RG's and KD's usual duties as forestry staff and were at the farmer's request based on prior complaints to RG about monkey damage to crops. The questionnaire we used to guide interviews is included in the Appendix A. We aggregated interview data to protect people's identities.

## 2.4. Research Permits and Approvals

Our research with langurs and with people was reviewed and approved by the UWICER Department of Forests and Park Services, Ministry of Agriculture and Forests (approval number DoFPS/Nga-5-35/6685804905F96956546240, dated 9 November 2020), Royal Government of Bhutan, and by Central Washington University's Institutional Animal Care and Use Committee (protocol 2019-111) and Human Subjects Review Board (protocol 2019-116).

## 2.5. Analysis

Using the statistical function in Microsoft Excel 2016 and in R, we analyzed data on golden langurs' group sizes and structures and sleep site data. We used inferential and

descriptive statistics to test our predictions. We calculated adult sex ratio and average group size in the BC and HS using Microsoft Excel. Using R, we set an alpha value (*p*) at ≤0.05 and performed *t* tests to determine whether there was a significant difference between the mean group sizes in the BC and HS.

## 3. Results

### 3.1. Golden Langur Group Sizes and Compositions in the HS and BC

We observed 24 golden langur groups consisting of 291 individuals. The mean sighting per each golden langur group was 10.33 times (N = 15 groups; range: 2–17 times) in the HS and 6.33 times (N = 9 groups, range: 2–11 times) in the BC. Overall average group size was $12.08 \pm 4.62$ individuals. Groups were significantly larger (t(22) = 0.01, $n_1$ = 9, $n_2$ = 15, $p < 0.05$) in the HS ($13.73 \pm 3.94$ individuals in the HS) compared to the BC ($9.55 \pm 3.04$ individuals), a result that supported our prediction. We observed multi-male/multi-female, uni-male/multi-female, and bi-male/multi-female group structures (Table 2).

**Table 2.** Social structures of groups living near a human settlement (HS) and in a biological corridor (BC), Langthel subdistrict, Trongsa district, central Bhutan.

| Group Type [1] | Human Settlement (Groups) | Biological Corridor (Groups) |
|---|---|---|
| Uni-male/multi-female | 4 | 5 |
| Bi-male/multi-female | 2 | 2 |
| Multi-male/multi-female | 9 | 2 |

[1] The uni-male/multi-female social structure is considered to be the most common and stable group formation type [13].

Groups living near the HS had a larger adult sex ratio, and 12/207 (6%) of the groups' individuals were infants/juveniles. Groups living in the biological corridor had a smaller adult sex ratio, a finding that supported our prediction. Ten out of eighty-four (12%) of the groups' individuals were infants/juveniles (Table 3).

**Table 3.** Golden langur age/sex compositions for groups living near a human settlement (HS) and in a biological corridor (BC) Langthel subdistrict, Trongsa district, central Bhutan.

| Age/Sex Class | Human Settlement (Individuals) | Biological Corridor (Individuals) | Overall (Individuals) |
|---|---|---|---|
| Adult male | 54 | 15 | 69 |
| Adult female | 96 | 42 | 138 |
| Subadult | 45 | 17 | 62 |
| Infant/juvenile | 12 | 10 | 22 |
| Adult sex ratio [1] | 0.36 | 0.26 | 0.33 |
| Adult:immature ratio | 0.38 (150:57) | 2.11 (27:57) | 0.41 (207:84) |

[1] Adult sex ratio = $N_{adult\ males}/(N_{adult\ males} + N_{adult\ females})$ [33].

### 3.2. Predator Surveys

We monitored 29 sleep locations visually and with camera traps: 21 in the HS and 8 in the BC. One sleep location (in the HS) was used more than once but all others were used only once in our dataset.

Mammals reported in Langthel subdistrict and ranging in both the BC and HS study areas included tigers (*Panthera tigris*), common leopards (*P. pardis*), clouded leopards (*Neofilis nebulosa*), Asiatic golden cats (*Catopuma temminckii*), dholes (*Cuon alpinus*), wild pigs (*Sus scrofa*), sambar deer (*Rusa unicolor*), barking deer (*Muntiacus muntjac*), alpine musk deer (*Moschus chrysogaster*), and red pandas (*Ailurus fulgens*), as well as Assamese macaques (*Macaca assamensis*) and golden langurs.

We recorded leopards and domestic dogs at 24% of sleep sites (N = 7/29) and raptors at 21% (N = 6/29) of sleep sites.

In the HS, we recorded two incidents of electrocution (one adult at a power transformer and one adult along the main power line; this latter individual was later found being eaten by a domestic dog) and one occurrence of a subadult male golden langur hit and killed by a vehicle. We also recovered the skull of a dead adult golden langur from a tree, which could have been leopard prey since leopards cache their prey in trees to avoid intraguild competition [34].

### 3.3. Characteristics of Sleep Sites

The golden langurs in our preliminary study exclusively slept in trees at night; we did not observe them sleeping in caves and/or on cliffs as has been reported for close relatives *Trachypithecus francoisi* [19] and *T. leucocephalus* [20]. We measured the characteristics of 76 sleep trees (Figure 5). All but one tree was used only once in our dataset. On one occasion, we observed the members of the group sleeping in the same tree crown, but usually the members of the group spread out to sleep across between two and five tree crowns (two trees, N = 10 observations; three trees, N = 13 observations; four trees, N = 3 observations; five trees, N = 2 observations).

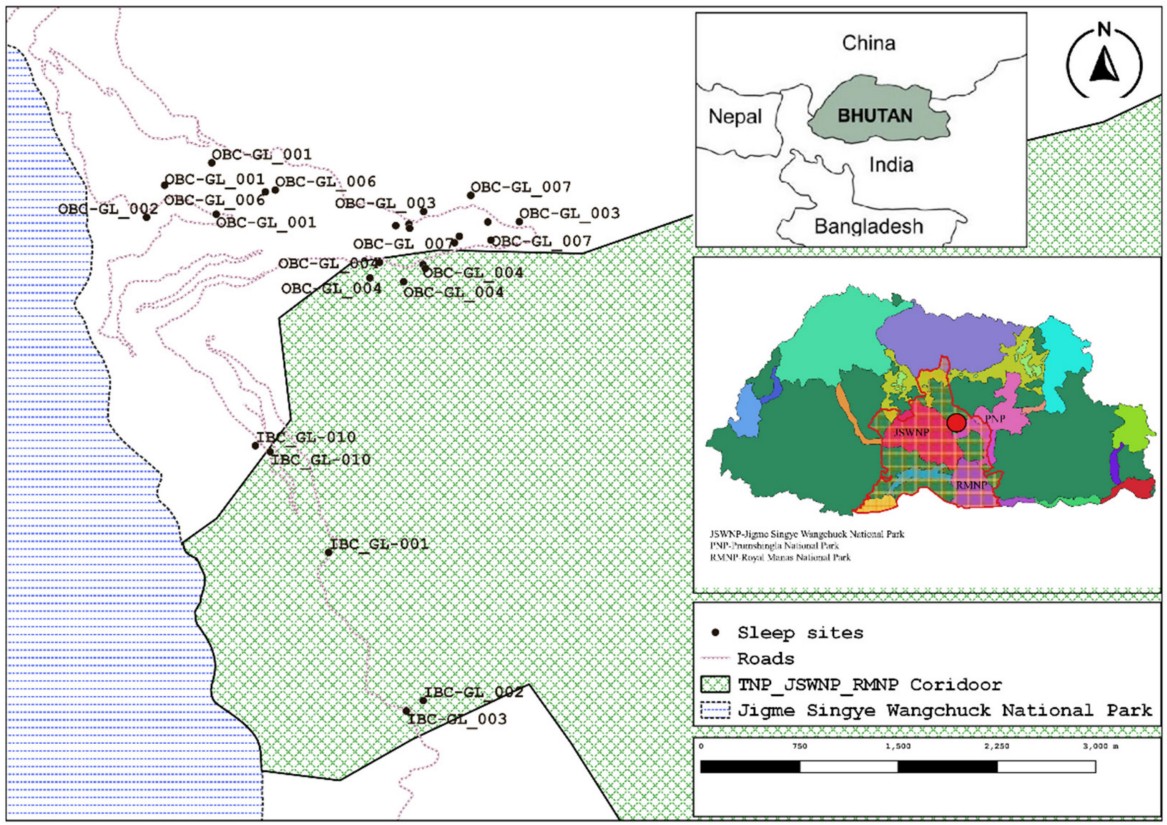

**Figure 5.** Locations of golden langur sleep trees in Langthel subdistrict, Trongsa district, central Bhutan.

The langurs most often slept on emergent trees (N = 38 trees) in the mid-level (N = 47 trees) of the canopy. Thirty-three (43.42%) sleep trees had low connectivity to the surrounding canopy. Sleep trees were usually rooted in areas with a strong (N = 14 trees, or 48%) or moderate (N = 8 trees, or 28%) slope; fewer sleep trees were rooted in ground with steep (N = 1 tree, or 3%) or gentle (N = 6 trees, or 21%) slopes. Most sleep sites were oriented south (N = 8 trees, or 28%), followed by southeast (N = 5 trees, or 17%), north (N = 4 trees, or 14%), and northwest (N = 2 trees, or 7%) orientations.

The 76 sleep trees were classified as 26 species of 20 families and 13 orders (Table 4). Four trees were not identified and eight were not identified to species level. The most frequently used species (26 of 76 trees) were *Sapium insigne* (N = 9 trees), *Altingia excelsa*

(N = 5 trees), and *Sapium eugeniigolium*, *Rhus succedanea*, and *Vitext pinnata* (N = 4 trees, respectively, in each species). Forty-five sleep trees (62.5%), including the three species most often used, had a spreading/oval shape. Sleep trees' mean DBH was 51.58 cm (SD = 30.82 cm, range: 12–160 cm) and the mean height was 19.37 m (SD = 7.50 m, range: 7.60–36 m). Golden langurs most often slept in trees of DBH range 26–50 cm (N = 38 trees) and height ranges of 11–15 m (N = 21 trees) and 16–20 m (N = 20 trees).

**Table 4.** Tree species used by golden langurs as sleep sites (N = 76 trees).

| Order [1] | Family | Species | Count |
|---|---|---|---|
| Apiales | Araliaceae | *Eluetherococcus* spp. | 1 |
| | | *Heteropanax chinenesis* | 1 |
| Boraginales | Boraginaceae | *Cordia obliqua* | 1 |
| Ericales | Theaceae | *Schima wallichii* | 3 |
| Fabales | Fabaceae | *Albizia* spp. | 3 |
| | Papilionaceae | *Erythrina* spp. | 2 |
| Fagales | Fagaceae | *Canstanopsis hystrix* | 3 |
| | | *C. tribuloides* | 1 |
| | | *Quercus griffithii* | 1 |
| | Juglandaceae | *Engelhardia spicata* | 1 |
| Lamiales | Lamiaceae | *Callicarpa arborea* | 2 |
| | | *Vitex pinnata* | 1 |
| Laurales | Lauraceae | *Cinnamomum glanduliferum* | 2 |
| Malpighiales | Euphorbiaceae | *Sapium insigne* | 9 |
| | | *S. eugeniigolium* | 4 |
| | Phyllanthaceae | *Bischopia javanica* | 3 |
| Malvales | Bombacaceae | *Bombax ceiba* | 2 |
| Myrtales | Combretaceae | *Terminalia* spp. | 2 |
| | Lythraceae | *Duabanga grandiflora* | 1 |
| | Myrtaceae | *Syzygium cumini* | 1 |
| Rosales | Moraceae | *Ficus tinctoria* | 3 |
| | Rosaceae | *Prunus cerasoides* | 1 |
| Sapindales | Anacardiaceae | *Rhus succedanea* | 4 |
| | | *Mangifera indica* | 1 |
| | Meliaceae | *Dysoxylum chisocheton* | 2 |
| Saxifragales | Altingiaceae | *Altingia excelsa* | 5 |
| Unknown | | | 4 |

[1] Classification based on [35–38].

## 4. Discussion

Data on Bhutan's golden langurs living outside of BCs and national parks are urgently needed because the majority of the population is found in these landscapes, which often have marked anthropic impacts [6,25]. We studied golden langurs living near an HS and in a BC in Trongsa district, central Bhutan. As has been documented at other sites in Bhutan and India [4,12,14], our golden langur groups living near the HS were significantly larger and included more adult males relative to adult females. Habitat quality and golden langur group sizes appeared to be inversely proportional: when forest habitat quality deteriorates, the number of individuals in each group increases [6]. A similar pattern has been noted in other golden langur studies: group sizes are larger when langurs live near people. In human-altered landscapes such as plantations and farms, this might occur due to a lack of dispersal options, increased reliance on crops, and an associated reluctance to disperse from natal groups, or in response to scramble competition. In their study of golden langurs in Assam, Shil and colleagues [12] found increasing average group sizes in forest-core (average 7.4 individuals per group), forest-edge (average 12.7 individuals per group), and plantation (average 13.9 individuals per group) sites. They noted that larger groups might occur in more fragmented habitats as a response to scramble competition. They also found that while group size did seem to be influenced by forest type, age/sex ratios within groups were not, perhaps because larger groups tended to include more adult males. In

their study of golden langurs in Bhutan, Lhendup and colleagues [4] found that average group sizes were significantly larger in disturbed (average 7.67 individuals per group) versus undisturbed (average 6.76 individuals per group) forests. Group compositions near HSs may reflect reliance on clumped and/or predictable distributions of nutrient-dense food resources (such as occurs in orchards or along roadways; see [39]), altered dispersal patterns compared to the natal group, a lack of dispersal options due to the scarcity of suitable habitats, and distinct predation patterns. The adult to immature ratio we calculated was low in the HS compared to the BC and compared to values calculated in Assam (0.6, 0.9, and 0.9 in forest core, edge, and plantation, respectively) [12] and Bhutan (2.12 and 1.86 in undisturbed and disturbed sites, respectively) [4]. Bhutan is perhaps distinct compared to Assam in terms of the presence of an intact predator array in both HSs and BCs, whereas predators may be hunted out or otherwise absent in disturbed, fragmented, or extensively altered habitats in Assam. The existence of predators, particularly those that prey on immature individuals, may explain the low numbers of immature individuals we observed in HS groups, coupled with other mortality risks associated with roads, powerlines, and areas where people use dogs to protect crops.

We recorded the presence of leopards, raptors, and, for groups living near the HS, domestic dogs near golden langur sleep sites. Leopards, dholes, and snakes such as pythons are documented predators of golden langurs [2], and raptors are common ambush predators of primates foraging on branches [40]. We tallied the frequencies of leopard signs, counted numbers of raptors hovering and within a 30 m radius of sleep sites, and recorded seven incidents of python encounters with local people, so predators were present at our study area; however, we saw no cases of predation on golden langurs. Langurs may be susceptible to predators when they are solitary, as occurs when adult males migrate between groups, when they are located toward the periphery of a group, or during periods of reduced vigilance, as occurs at sleep sites.

Domestic dogs are also golden langur predators and may target or have greater hunting success with smaller, immature individuals; for example, Chetry and co-workers [24] report seven golden langur deaths from domestic dogs in one year, of which five were juvenile langurs. We did not observe any golden langur mortality directly attributable to domestic dogs, but we did observe a domestic dog eating a golden langur that had been electrocuted on powerlines. Conversely, natural predators such as leopards may be deterred from hunting near HSs, although farmers reported to us the presence of leopards in or near fields, and one farmer described to us how a leopard killed his adult dog, which the farmer had formerly relied on to drive langurs from his orchard, and we recovered the skull of an adult golden langur from a tree that may have been a leopard prey cache site.

The tree species *Sapium insigne* (synonym of *Falconeria insignis*, classified in the spurge family Euphorbiaceae) appears to be particularly important to conserve in golden langur habitats, as this tree was most often used for sleeping in our dataset. Additionally, langurs most often used emergent trees rooted in ground with strong or moderate slopes and oriented south or southeast, perhaps because these trees provided good visibility of the surrounding area. Langurs most often slept in the mid-region of the crown in trees that were 11 m tall or taller with girths exceeding 26 cm. Individuals of the langur group usually spread among several, most often three, neighboring trees to sleep, but once we saw the entire group sleep in the crown of one tree.

Anecdotally, we think that golden langurs prefer *Sapium insigne* as sleeping sites because this species has spreading and open crown shapes and can accommodate almost all family members in the same tree crown. Golden langurs living near human settlements may prefer *Sapium insigne* as people do not consider these species good for timber, so these trees are rarely cut down by local people. In the BC, our anecdotal data indicated that golden langurs also prefer to sleep in *Calicarpa arborea* and *Schima wallichiana* trees, both of which are timber species used by local people, but their extraction is regulated by policies in force in protected areas. We were not able to test for sleep site preferences because we had not yet assessed the distribution of tree species in the BC and near the HS; we collected

a few sleep-site data points and our data collection spanned one season due to COVID-19 restrictions on travel.

All sleep sites we recorded were trees. One sleep tree was reused in our study period (of the species *Sapium insigne*), but other sleep sites appeared to be single-use. This infrequent reuse of sleep sites has been taken by previous researchers as evidence of predation avoidance being the primary driver of sleep site selection, as opposed to access to food resources or other causes [19,20]. At Nonggang Nature Reserve (China), *Trachypithecus francoisi* exclusively slept on cliffs (23 sleep sites; 17 ledges and 6 caves) and reused seven of these sites more than nine times each [19]. Similarly, *T. leucocephalus* in Fusui Nature Reserve (China) used 18 sleep sites (1 ledge and 17 caves), and all sleep sites were reused more than once [20]. Our results should be interpreted with caution as the COVID 19 pandemic shortened our field season and meant that our dataset spanned late fall to early spring. Golden langur ranging patterns and sleep sites are likely impacted by season. Although we lacked sufficient data to compare the characteristics of sleep trees near the HS with those in the BC, our preliminary data indicated that predation, which included natural predators and domestic dogs near the HS, drove sleep site selection at both locations. Although cliffs and caves exist in Trongsa district in both the BC and HS, we did not observe golden langurs using them as sleep sites in our dataset; however, two of us (RG and KD) have observed langurs on cliffs during the day using them as salt licks (Figure 6), so it seems likely that additional study will show golden langurs' use of cliffs and caves, too.

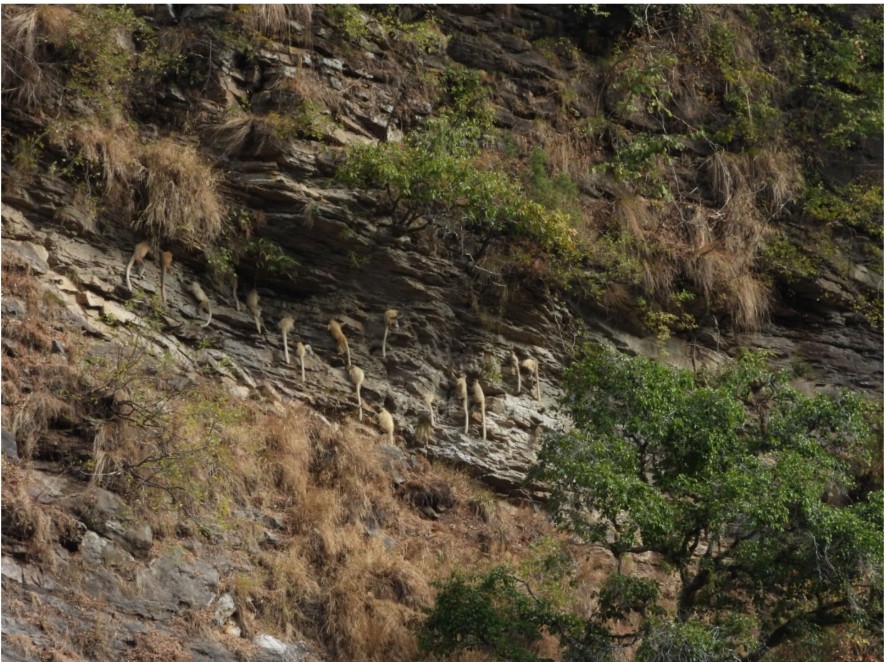

**Figure 6.** Photo showing golden langurs at a mineral lick site.

In addition to death due to predation, golden langurs living near people experience mortality due to electrocution on power lines and vehicle collisions [6], which increases the costs golden langurs experience from living near people [39]. We observed both causes of mortality during the short duration of our study. If this mortality is biased toward a particular sex or age class, it could contribute to the differences observed in the group composition of langurs living in anthropic versus less disturbed habitats. In Trongsa district, golden langurs are often spotted on or near roadways, and they sometimes cross roads on the ground when the canopy is bisected by a roadway. At higher elevations, the use of salt to melt roadway ice attracts grey langurs (*Semnopithecus schistaceus*) to the road as a salt lick, and at lower altitudes, such as our study site, on cool or cold days,

golden langurs sometimes bask in the sun on the warm asphalt. Increased use of signage and installation of speed bumps and insulated wires would greatly reduce golden langur mortality for those populations living near anthropic environments.

Thinley and colleagues [25] noted that, at some sites, farmers engage in retaliatory killing of crop-reliant golden langurs. We did not observe this cause of golden langur mortality during our study, but we note that subsistence farmers in Bhutan are increasingly vulnerable to food insecurity due to temperature and rainfall changes associated with climate change [41]. Currently, farmers are adapting to these changes by increasing their amounts of off-farm labor and by working on others' farms to supplement income, but one study recorded increased wild collection and livestock grazing to supplement what is produced on farms [41]. This precarious farming situation and people's behavioral adaptations seem likely to increase golden langurs' reliance on crops and encounters between langurs and people. Our ongoing research focuses on assessing costs and benefits for people [26] and for golden langurs [39] living in close association with people.

## 5. Conclusions

We found that golden langurs living near human settlements changed their sleeping sites every night. We observed them sleeping in tall trees with large girths and spreading branches. Tall trees may be preferred to avoid predators and because all group members can sleep in the crown of one tree if its branches are spreading. We recommend continued enforcement of regulations limiting timber extraction of *Sapium insigne*, *Calicarpa arborea*, and *Schima wallichiana* as these tree species appear to be golden langurs' preferred sleep sites. We also advise a golden langur community awareness campaign for foresters and local people.

**Author Contributions:** Conceptualization, K.D. and L.K.S.; methodology, K.D. and R.G.; formal analysis, K.D. and T.E.; investigation, K.D. and R.G.; resources, K.D. and T.E.; data curation, K.D. and L.K.S.; writing—original draft preparation, K.D. and L.K.S.; writing—review and editing, K.B., T.E., R.G., and N.P.D.; visualization, N.P.D. and K.D.; supervision, L.K.S. and K.B.; project administration, K.D.; funding acquisition, K.D. and L.K.S. All authors have read and agreed to the published version of the manuscript.

**Funding:** This research was supported by the Rufford Foundation (K.D.); the School of Graduate Studies and Research, Central Washington University (K.D.); and Primate Conservation, Inc. (L.K.S.).

**Institutional Review Board Statement:** The study was conducted according to the guidelines of the Declaration of Helsinki, and approved by the Institutional Review Board of Central Washington University (IACUC protocol code 2019-111 and HSRC protocol code 2019-116, both approved on 14 October 2019).

**Informed Consent Statement:** Informed consent was obtained from all subjects involved in this study, following the protocol approved by Central Washington University (HSRC protocol code 2019-116 approved on 14 October 2019).

**Data Availability Statement:** The data presented in this study are available on request from the corresponding authors.

**Acknowledgments:** We thank the Ugyen Wangchuck Institute for Conservation and Environmental Research (UWICER) and the Department of Forests and Park Services for granting research clearance. Thank you to Jigme Singye Dorji, Class 10, Jakar High School, and Kinzang Sonam, class 11, Dechentsemo Central School, for assisting us in data collection.

**Conflicts of Interest:** The authors declare no conflict of interest.

## Appendix A

Local people's interviews on perceptions of and attitudes towards Golden Langurs, Langthel subdistrict, central Bhutan.

**Table A1.** Respondent detail.

| | |
|---|---|
| Respondent ID **: | Site/Location: |
| Gender: M/F      Age (18–25), (26–40), (41–55), (55 and above) | **Occupation:** |
| Education (Monastic, NFE, Primary/High School/Graduate) | |

** ID can be written as day/date/year/respondent number (e.g., 19/03/2020/01).

**Table A2.** People's knowledge on golden langurs.

| Questions | Responses |
|---|---|
| Tell us about golden langurs (what they look like, size (male and female), color) | |
| Have you seen golden langurs? If yes, what were they doing at the time of sighting? | |
| How many of them were there? What were they eating? | |
| Where do they mostly sleep? | |

**Table A3.** People and golden langur interactions.

| Questions | Responses |
|---|---|
| What do you do when you see them? | |
| When (season) do golden langur visit your farm? | |
| How often they visit your farm? | Daily; once in 3 days; once in a week; once in a while; never |
| What crops or features of your farm do golden langur mostly damage? | |
| What is the extent of that damage? | |
| What do you do to keep golden langur away from your farm? | |
| Have you altered your farming practices in the last five years, and if so, why? | |
| Can you cite one example of the changes in your farming (e.g., rice cultivation to vegetables) | |

**Table A4.** Perceived threats for golden langurs.

| Questions | Responses |
|---|---|
| If Bhutanese rules and policies permit, what would you opt to do with problematic langur groups? | |
| If Bhutanese rules and policies permit, would you like to have a golden langur as a pet? | |
| Do you still practice shifting cultivation? | |
| Can you think of any uses for golden langur pelts? | |
| Do you think the golden langur population has changed trend over the past 5 years? | Decreased, remained stable, or increased |
| Do you feel golden langur are beneficial to you? If yes, how? | |
| Are you aware that the golden langur is an endangered species? | **Yes/NO** |
| Do you know that golden langurs are listed schedule I animals in the FNCR 2017? | **Yes/No** |

**Table A5.** Predation of golden langurs.

| Questions | Responses |
|---|---|
| Have you encountered any large cats in your locality? If yes, what species? | |
| Have you encountered python snakes in your locality? If yes, where? | |
| Did you see raptor birds in your locality? If yes, what species? | |
| Do you have dogs? If yes, how many? If no, does your neighbor have dogs? If yes, how many? | |

**Table A6.** Cultural significance of golden langur.

| Questions | Responses |
|---|---|
| Are golden langurs associated with any cultural beliefs or religious significance? | |
| Why is an encounter with a golden langur considered a good omen? Explain. | |
| Relate a story involving a golden langur and four friends. | |

**Table A7.** Comments, if any.

| Questions | Responses |
|---|---|
| General comments (suggestion, recommendations) | |

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
