# Peer review of "Preliminary Report on Golden Langur (Trachypithecus geei) Winter Sleep Sites"

_humans, doi:10.3390/humans1020005_

Round 1

Reviewer 1 Report

This is an interesting paper written on the sleep sites of Golden Langur. I have the following comments to improve the paper.

  • The abstract and introduction are well written.
  • Materials and methods section
  • How did you come up with 24 groups of golden langurs for this study? Is it only based on the experience of one of the authors or is there any other supporting evidence for this?
  • How did you collect data and analyze them? You can briefly mention the data collection procedure and analysis part as well. Also, you can put the list of a questionnaire and supportive database in the annex section. The questionnaires for both parts; sleep behavior, and predators’ presence survey would be helpful. I believe you must have approved such questionnaires from University’s review board during the approval process as well.
  • You have mentioned you used scan sampling both in abstract and methods but it is not described at all. Can you elaborate a bit on it and the rationale behind choosing this mode of sampling?
  • In the predator presence survey, you used the points counts method with a fixed radius of 30 m and 15-minutes stops to document the presence of raptors? Can you explain why you choose this particular method and how did you capture the data that doesn’t fall within the fixed radius of 30 m.

  • Results
  • You assumed 24 Golden langur groups before the study and got the exact same result after your survey as well, right? It is interesting to note that there is a higher number of langur groups in HS than in BC. I think you should discuss it both in the results and discussion section with a little bit more clarification on it.
  • On predator survey, the paper just lists the name of predators but doesn’t explain how the langurs fall into the predators. I think it would be helpful if you could explain a bit about it as well.
  • Discussion and Conclusion
  • Sapium insigne was found a favorite tree for golden langurs. Are there any particular reasons for this? Also are there any particular differences in terms of preference between the choice of species in HS and BC?
  • Maybe the conclusion part is missing in the paper. If you could provide a brief conclusion with key take-home messages, that would be great.

Author Response

Thank you for your careful consideration of our work and helpful comments on the manuscript. We incorporated all suggestions into the revised manuscript (visible with track changes) and include a point-by-point response to each reviewer below.

We have added emphasis to the fact that the golden langurs we studied are impacted by living near humans, and humans are affected by the langurs. Please see, for example, LL 309-323, 356-368, and 392-406.

Sincerely,

Lori K. Sheeran and Kuenzang Dorji, on behalf of all co-authors, all of whom also reviewed this response

Reviewer 2 Report

The research aim, methods and results are clear. It provides a preliminary description on the ecology and behaviors, especially the sleep sites of golden langurs in Bhutan. It has a contribution to the understanding of golden langurs.

My question is: (1) what is the implications of studying the sleep sites and langur-predator interactions? (2) what is the contribution of this study to previous studies on (golden) langurs? (3) it is better to provide a photo of the field and golden langurs.

Author Response

(The authors gave the same response as above.)

Reviewer 3 Report

This a clearly presented, well-written preliminary report, essentual to the preservation of the environment of these creatures. Unfortunately, it is not presented within the context of topics relevant to human evolution/human variation.  I am not saying it should, simply that it does not.  That being the case, I think it would be better for the efforts of the authors and growth of this new journals for this work to be published in a different MDPI venue. 

Author Response

(The authors gave the same response as above.)

Round 2

Reviewer 3 Report

My thanks to the authors for addressing my concerns as well as those of the other reviewers.